# Comparing SARS-CoV-2 Antibody Responses after Various COVID-19 Vaccinations in Healthcare Workers

**DOI:** 10.3390/vaccines10020193

**Published:** 2022-01-26

**Authors:** Yu-Kyung Kim, Dohsik Minn, Soon-Hee Chang, Jang-Soo Suh

**Affiliations:** 1Department of Clinical Pathology, School of Medicine, Kyungpook National University, Daegu 41944, Korea; kimyg@knu.ac.kr (Y.-K.K.); marta10@hanmail.net (S.-H.C.); 2Department of Diagnostic Immunology, Seegene Medical Foundation, Seoul 05548, Korea; dsmin@mf.seegene.com

**Keywords:** COVID-19 vaccine, AstraZeneca-Oxford, Pfizer-BioNTech, neutralizing antibody, Elecsys Anti-SARS-CoV-2 S assay, Abbott SARS-CoV-2 IgG II Quant assay

## Abstract

Coronavirus disease 2019 (COVID-19) vaccination began for healthcare workers in South Korea at the end of February 2021. This study investigated severe acute respiratory syndrome coronavirus 2 (SARS-CoV-2) antibody responses after various COVID-19 vaccinations in healthcare workers. Blood specimens of 497 vaccinated healthcare workers were collected. Inoculated vaccines were ChAdOx1 (AstraZeneca/Oxford), BNT162b2 (Pfizer/BioNTech), JNJ-78436735 (Janssen), and mRNA-1273 (Moderna). Each specimen was tested for antibodies against SARS-CoV-2 using Elecsys Anti-SARS-CoV-2 S assay (Roche Diagnostics), SARS-CoV-2 IgG II Quant assay (Abbott), and R-FIND SARS-CoV-2 Neutralizing Antibody kit (SG medical Inc.). A questionnaire was used to investigate adverse events related to vaccination. We found that 99.5% of the subjects showed a 96–100% positive rate in all three antibody assays, regardless of the vaccine type. The antibody-positive rate of completed vaccination groups reached 96–100%, and antibody quantities significantly increased 2 weeks after vaccination. The antibody values measured approximately 3 months after BNT162b2 inoculation significantly correlated with adverse events.

## 1. Introduction

After the onset of global coronavirus disease 2019 (COVID-19) pandemic, vaccines against severe acute respiratory syndrome coronavirus 2 (SARS-CoV-2) were developed, and vaccination was started worldwide [1]. In South Korea, COVID-19 vaccination began at the end of February 2021, for 60,000 healthcare workers related to COVID-19 as a priority. The vaccines introduced were ChAdOx1 (AstraZeneca/Oxford), BNT162b2 (Pfizer/BioNTech), JNJ-78436735 (Janssen), mRNA-1273 (Moderna), and NVX-CoV2373 (Novavax). mRNA vaccines, such as BNT162b2 and mRNA-1273, reportedly have an efficacy of 95% [2,3] and are used in several countries at present. Although the efficacy of the viral vector vaccine (ChAdOx1 and JNJ-78436735) was reported to be lower than that of the mRNA vaccine, it met the minimum efficacy standards of the World Health Organization and was used for early-stage vaccination in 2021 [4].

At present, various immunoassays for SARS-CoV-2 antibody detection are available, including enzyme-linked immunosorbent assay (ELISA) and chemiluminescence enzyme immunoassays [5,6]. To evaluate the effectiveness of vaccination, quantitative antibody detection should be conducted. In this regard, several studies have been conducted on the antibody figurequantities or antibody positive rate after vaccination [7,8,9,10,11,12]. However, vaccination and the SARS-CoV-2 antibody detection test are uncommon in real-world clinical practice. Therefore, it is necessary to conduct various studies and evaluate the effectiveness of vaccines to determine future vaccine policies. In this study, we measured the quantities of SARS-CoV-2 antibodies after vaccination in health care workers and analyzed the differences according to vaccine type and the elapsed days after vaccination. In addition, we investigated adverse reactions after vaccination through questionnaires. Lastly, we investigated the relationship between adverse events and antibody quantities. 

## 2. Materials and Methods

### 2.1. Subjects

This study included 497 volunteer healthcare workers who received a COVID-19 vaccine in Kyungpook National University Hospital between March and July 2021. Subjects previously infected with SARS-CoV-2 were not included. The inoculated vaccines were ChAdOx1 (AstraZeneca/Oxford, Lund, Sweden), BNT162b2 (Pfizer/BioNTech, Philadelphia, PA, USA), JNJ-78436735 (Janssen, NJ, USA), and mRNA-1273 (Moderna, MA, USA) (Table 1). Blood was collected in mid-July 2021, without considering the elapsed days after vaccination. After blood collection, the serum was refrigerated, and antibody tests were performed within 2 days. To investigate the adverse events related to vaccination, we conducted a questionnaire consisting of the following grades: Grade 1, no discomfort at all; Grade 2, discomforts but no problems in daily life; Grade 3, symptoms requiring self-medication; Grade 4, symptoms requiring outpatient treatment; and Grade 5, symptoms requiring hospitalization. Regardless of the type of adverse event, subjects were graded according to the degree of need for treatment.

### 2.2. SARS-CoV-2 Antibody Measurement 

Every blood specimen was tested for antibodies against SARS-CoV-2 using Elecsys Anti-SARS-CoV-2 S electrochemiluminescence immunoassay (Roche Diagnostics GmbH, Mannheim, Germany), SARS-CoV-2 IgG II Quant assay (Abbott, Chicago, IL, USA), and R-FIND SARS-CoV-2 Neutralizing Antibody ELISA kit (SG medical Inc., Seoul, Korea).

#### 2.2.1. Elecsys Anti-SARS-CoV-2

The Elecsys Anti-SARS-CoV-2 S assay is for the quantitative detection of antibodies (including IgG) against SARS-CoV-2 spike (S) protein receptor-binding domain (RBD). According to the manufacturer’s instructions, we conducted this assay using the Cobas e 801 analyzer (Roche Diagnostics). An antibody concentration of <0.80 U/mL was considered negative, and that of ≥0.80 AU/mL was considered positive. The measuring range spanned from 0.4 U/mL to 2500.0 U/mL.

#### 2.2.2. Alinity SARS-CoV-2 IgG

The SARS-CoV-2 IgG II Quant assay is a chemiluminescent microparticle immunoassay, which is used for the qualitative and quantitative determination of IgG antibodies to SARS-CoV-2, using the Alinity i system (Abbott). This assay can detect IgG antibodies, including neutralizing antibodies, to the receptor RBD of the S1 subunit of S protein of SARS-CoV-2. A concentration of <50 AU/mL was considered negative and ≥50 AU/mL was considered positive (measuring range: 21–40,000 AU/mL).

#### 2.2.3. R-FIND SARS-CoV-2 Neutralizing Antibody

An R-FIND SARS-CoV-2 ELISA kit was used for the qualitative measurement of neutralizing antibodies against SARS-CoV-2 in human serum using competitive ELISA. The assay results were interpreted as follows: the average value of optical density (OD) of the negative control (Avg. of NC) was calculated. Next, the inhibition rates were calculated using the following formula: Signal Inhibition = (1 − OD value of Sample/Avg. of NC) × 100%. The assay results were determined as positive (measured value ≥ 30% signal inhibition) and negative (measured value < 30% signal inhibition).

### 2.3. Ethical Approvals

This protocol of the study was reviewed and approved by the Kyungpook National University Hospital Institutional Review Board (approval No. 2021-03-037). Participants were informed of the study title, purpose, and protection of personal information at the beginning of the study. They were notified that their participation was voluntary and that they could withdraw from the study at any time.

### 2.4. Statistical Analysis

One-way analysis of variance or Kruskal–Wallis test was used to compare the difference in antibody quantities according to the elapsed day of subjects who completed the ChAdOx1 vaccination. The independent *t*-test was used to compare the antibody quantities of the adverse-event group and no-discomfort group. All statistical analyses were performed using GraphPad Prism version 9.0.0 for Windows (GraphPad Software, San Diego, CA, USA) and SPSS software ver. 27.0 (IBM, Armonk, NY, USA).

## 3. Results

### 3.1. Demographics of Subjects

Demographics and vaccination information of the subjects are shown in Table 1. The study comprised individuals in their 20 s and 50 s, who accounted for 26.4% and 27.0%, respectively, of the study subjects, and 80.9% of the 497 subjects were female. Furthermore, 79.9% (397/497) of the subjects completed vaccination. Subjects who completed ChAdOx1 and BNT162b2 vaccination accounted for 61.0% (303/497) and 17.7% (88/497), respectively. The elapsed days after the last vaccine administration ranged from a minimum of 0 days to a maximum of 117 days (Table 2).

### 3.2. Antibody-Positive Rate and Quantity in Each Assay

#### 3.2.1. Results of Elecsys Anti-SARS-CoV-2 Assay

The antibody positivity rate was 100% in the group that completed vaccination, regardless of the vaccine type or elapsed day after the final injection (Table 2). The elapsed time periods after the final ChAdOx1 and JNJ-78436735 injections were 29.3 ± 9.1 days and 22.8 ± 2.1 days, respectively, and the quantities of antibody were 1017.4 ± 718.4 IU/mL and 15.2 ± 17.4 IU/mL, respectively. Among the incomplete vaccination group, the antibody-positive rate of mRNA-1273, which had a mean elapsed time of 9.9 days (standard deviation (SD): 4.0 days) after vaccination, was 56.5%. Although vaccination was incomplete in the case of ChAdOx1, which had a mean elapsed period of 66.2 days (SD: 31.3 days) after the first injection, the antibody-positive rate was 100%.

#### 3.2.2. Results of Alinity SARS-CoV-2 IgG

In the ChAdOx1, BNT162b2, and JNJ-78436735 vaccine completion groups, antibody positivity was 99–100% (Table 2). The antibody-positive rate of mRNA-1273, which had a mean elapsed time of 9.9 days (SD: 4.0 days) after vaccination, was 56.5%. One of the two subjects who received mixed vaccines (ChAdOx1/BNT162b2) was antibody negative. In the incomplete vaccination group, antibody positivity was 58.1% and 91.0% in mRNA-1273 and ChAdOx1, respectively.

#### 3.2.3. Results of R-FIND SARS-CoV-2 Neutralizing Antibody

Among the completed vaccination group, antibody positivity rates in the ChAdOx1, BNT162b2, and JNJ-78436735 vaccine completion groups were 96%, 100%, and 100%, respectively (Table 2). However, the antibody-positive rate of the mixed vaccines (ChAdOx1/BNT162b2) group was 0% (0/2). Moreover, antibody positivity rates in the mRNA-1273 and ChAdOx1 vaccine incompletion groups were 53.2% and 68.2%, respectively.

### 3.3. Antibody Quantities According to the Elapsed Day

In 303 subjects who completed the ChAdOx1 vaccination, the difference in antibody amount was investigated for each elapsed period after the final injection (Figure 1). There was no significant difference in antibody quantities in Elecsys Anti-SARS-CoV-2 assay over the entire period. For the Alinity SARS-CoV-2 IgG assay, compared with the amount of antibody at 0–14 d, it increased significantly on days 36–42, and there was no significance in the remaining period. We also found that the R-FIND SARS-CoV-2 neutralizing antibody was significantly increased for all periods after day 15, compared with the antibody quantities on days 0–14. Although there was no statistical significance, the amount of antibody decreased after 43 days in all three antibody test results.

### 3.4. Incidence of Adverse Events According to Vaccine Type

Figure 2 shows the adverse event results of 391 subjects who completed the vaccination (303 of the ChAdOx1, 88 of BNT162b2). Among the 303 participants who received the first ChAdOx1 dose, 156 (51.5%) reported a mild adverse event requiring self-medication (Grade 3), and 14 (4.6%) reported symptoms requiring clinical treatment (Grade 4). After the second ChAdOx1 dose, Grade 3 and 4 symptoms were found in 12.2% (37/303) and 2.3% (7/303) subjects, respectively. Eighty-eight subjects completed the BNT162b2 vaccination, with 11.4% (10/88) reporting Grade 3 symptoms and 1.1% (1/88) reporting Grade 4 symptoms after the first dose. However, after the second BNT162b2 dose, 41 (46.6 %) reported Grade 3 symptoms and 4 (4.5%) reported Grade 4 symptoms. In addition, 1/88 subjects who completed the BNT162b2 reported hospitalization (Grade 5) after the first dose. This was the only case of Grade 5 adverse events among the 497 subjects.

### 3.5. Association between Antibody Quantities and Adverse Events

Among the vaccine completion groups of ChAdOx1 and BNT162b2, 277 and 83 subjects, respectively, who had passed 15 days or more were selected (Table 3). Subjects who experienced Grade 3 or higher symptoms at the first or second dose were classified into the adverse-event group, whereas the remaining subjects were classified into the no-discomfort group. 

#### 3.5.1. Results of ChAdOx1

The age of the adverse event group was significantly lower than that of the no discomfort group (*p* < 0.01). In all three antibody tests, the antibody quantities were not significantly different between the two groups.

#### 3.5.2. Results of BNT162b2

In all three antibody tests, the antibody quantities of the adverse event group were higher than those of the no discomfort group. In particular, the antibody titer measured by the Alinity SARS-CoV-2 IgG test was statistically significantly higher in the adverse event group than in the no discomfort group (Table 3).

## 4. Discussion

In this study, 395 (99.5%) of the 397 subjects who completed COVID-19 vaccination showed 96–100% positive rate in all three antibody assays, regardless of vaccine type. The other two (0.5%; SARS-CoV-2 antibody negative for neutralizing antibody assay) were within 4 days of the last vaccination. When the vaccine was injected only once, the seropositivity rates of ChAdOx1 and mRNA-1273 were 68.2–100% and 53.2–56.5%, respectively. In the ChAdOx1 group, an average of 66.2 days (SD: 31.3 days, range: 7–117 days) elapsed after the first dose, whereas in the mRNA-1273 group, an average of 9.9 days (SD: 4.0 days, range: 4–18 days) elapsed. Further, it was thought that the seropositivity rates of mRNA-1273 were lower because the number of elapsed days after vaccination was insufficient. For example, Jeong et al. reported that the antibody quantity after the first ChAdOx1 dose was significantly higher at 21–28 days than at 11–20 days [7]. Similar reports showed that the antibody titer after two weeks was significantly higher than that after one week, even when the second vaccination was completed [6]. In various studies that evaluated the antibody response after COVID-19 vaccination, blood was collected at least two weeks after vaccination to measure antibodies [9,10,11,12,13]. In addition, we found that the SARS-CoV-2-neutralizing antibody was significantly increased for all periods after day 15 compared with the antibody quantities on days 0–14 in 303 subjects who completed the ChAdOx1 vaccination. We also found contrasting differences in symptoms between the two vaccines. In the ChAdOx1 vaccine group, Grade 3–5 symptoms were 56.1% after the first dose and decreased to 14.5% after the second dose. In the BNT162b2 group, Grade 3–5 symptoms increased from 13.6% (first dose) to 51.1% (second dose), showing a pattern opposite to that of ChAdOx1. Hwang et al. also showed that local and systemic adverse events were comparable after the first ChAdOx1 dose and the second BNT162b2 dose [14]. In this study, the association between adverse events and antibody quantities differed depending on the vaccine type. In the ChAdOx1 group, a significant association was noted between adverse events and age, but not with antibody quantities. Regarding BNT162b2, the antibody quantities were higher in the adverse-event group. Whether these differences were caused by the type of vaccine could not be definitively concluded. The distribution of days after vaccination using ChAdOx1 (mean ± SD: 30.1 ± 7.9 days) was higher than that using BNT162b2 (mean ± SD: 94.7 ± 2.2 days). Therefore, the effect of the lapsed date could not be excluded with certainty. Hwang et al. measured antibody quantities 2 weeks after vaccination and concluded that the immunogenicity was not associated with adverse events [14]. In this study, the antibody values, measured at approximately 3 months after BNT162b2 inoculation, significantly correlated with adverse events.

Furthermore, there has been much research evaluating the performance of various SARS-CoV-2 antibody assays. Notably, quantitative analysis is essential to evaluate the effectiveness of a vaccine, so evaluation of several recently developed quantitative analysis methods is being actively performed [7,8,9,10,11,12,13]. We used three assays with different target antibodies: (1) antibodies (including IgG) against SARS-CoV-2 S protein RBD; (2) IgG antibodies (including neutralizing antibodies) against the RBD of the S1 subunit of the SARS-CoV-2 S protein; and (3) neutralizing antibodies. As there are differences in the target antigen, measurement method, and positive cut-off criteria for each assay, our results showed differences in the antibody positivity rate of the same subject. In the ChAdOx1 (1st dose) group, positivity rates of the three assays were 100%, 91.0%, and 68.2%, respectively (Table 2). We measured antibody values without extending the measuring range of all three assays. Compared with other assays, the measurement range of the Elecsys Anti-SARS-CoV-2 is narrow, and many results exceeding the upper limit were included.

This study has some limitations. First, we did not strictly control the elapsed period after vaccination. Therefore, it is necessary to consider the number of days elapsed after inoculation when interpreting the results of this study. Second, the precise antibody amount could not be calculated for some samples exceeding the detection limit. As a result, we could not compare the concordance between the three SARS-CoV-2 antibody assays. Another limitation is that the samples (up to 4 months after vaccination) included in this study alone cannot be used to evaluate the long-term efficacy of the vaccine.

## 5. Conclusions

This study reported that the antibody-positive rate of completed vaccination groups reached 96–100%, and the antibody quantities significantly increased two weeks after vaccination. The postvaccination side effects were higher after the first ChAdOx1 dose and second BNT162b2 dose. The antibody values measured at approximately 3 months after BNT162b2 inoculation significantly correlated with adverse events.

## Figures and Tables

**Figure 1 vaccines-10-00193-f001:**
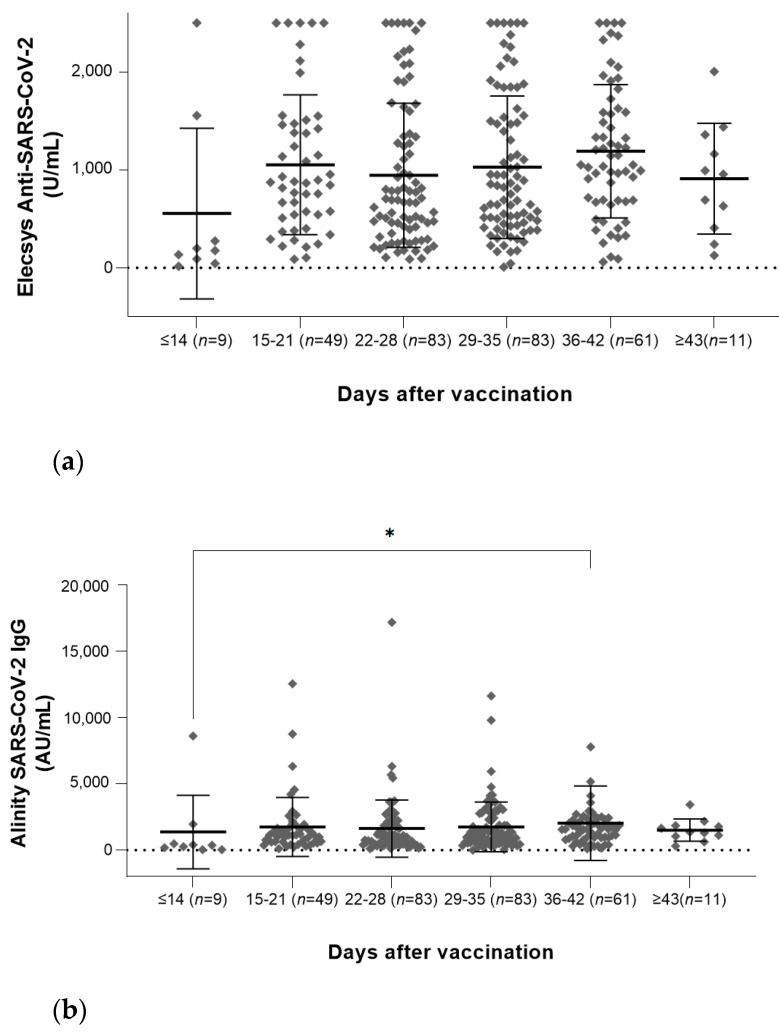
Antibody quantities according to the elapsed time period after completion of vaccination using (**a**) Elecsys Anti-SARS-CoV-2 assay, (**b**) Alinity SARS-CoV-2 IgG assay, and (**c**) R-FIND SARS-CoV-2 neutralizing antibody. In 303 subjects who completed the ChAdOx1 vaccination, the difference in antibody amount was investigated for each elapsed time period after the final injection. (* *p* < 0.05; † *p* < 0.01).

**Figure 2 vaccines-10-00193-f002:**
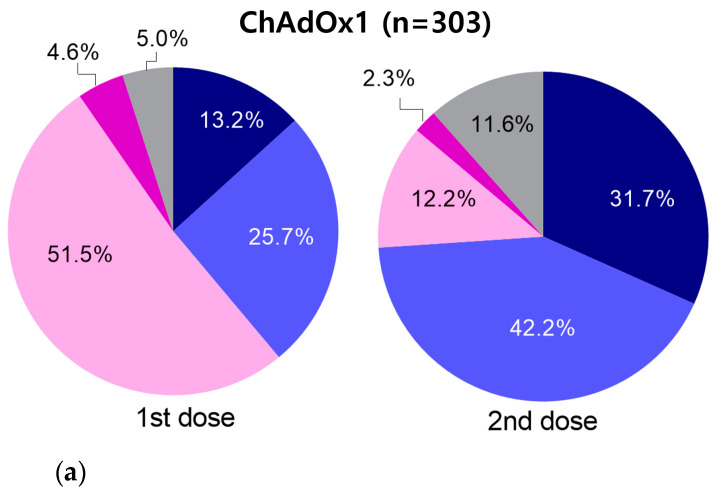
Differences in adverse events depending on the type of vaccine. Adverse events were assessed for 391 subjects who completed the vaccination ((**a**) 303 of the ChAdOx1 group, (**b**) 88 of the BNT162b2 group).

**Table 1 vaccines-10-00193-t001:** Demographics of the total of 497 subjects.

Characteristic	Number	%
Age		
20–29	131	26.4
30–39	87	17.5
40–49	108	21.7
50–59	134	27.0
≥60	31	6.2
Not available	6	1.2
Sex		
Female	402	80.9
Male	93	18.7
Not available	2	0.4
Vaccination		
Completion		
ChAdOx1 (1st and 2nd dose)	303	61.0
BNT162b2 (1st and 2nd dose)	88	17.7
JNJ-78436735 (1st dose)	4	0.8
ChAdOx1 (1st dose) and BNT162b2 (2nd dose)	2	0.4
Incompletion		
mRNA-1273 (1st dose)	62	12.5
AZD1222 (1st dose)	22	4.4
BNT162b2 (1st dose)	1	0.2
Not available	15	3.0

**Table 2 vaccines-10-00193-t002:** Antibody-positive rates and quantities of antibodies in each vaccination group.

	Number	Days after the Final Vaccination * (Range)	Elecsys Anti-SARS-CoV-2		Alinity SARS-CoV-2 IgG		R-FIND SARS-CoV-2 Neutralizing Antibody	
			Number of Positive Cases	Antibody Quantities (U/mL) *	Number of Positive Cases	Antibody Quantities (AU/mL) *	Number of Positive Cases	Neutralization Rates (%) *
Completed vaccination (N = 397)								
ChAdOx1 (1st and 2nd dose)	303	29.3 ± 9.1 (0–58)	303 (100%)	1017.4 ± 718.4	300 (99.0%)	1742.6 ± 2205.4	291 (96.0%)	81.6 ± 19.5
BNT162b2 (1st and 2nd dose)	88	94.7 ± 2.2 (88–102)	88 (100%)	948.5 ± 533.9	88 (100%)	3558.9 ± 2308.0	88 (100%)	90.8 ± 7.0
JNJ-78436735 (1st dose)	4	22.8 ± 2.1 (20–25)	4 (100%)	15.2 ± 17.4	4 (100%)	335.4 ± 324.2	4 (100%)	43.3 ± 25.1
ChAdOx1 (1st dose) and BNT162b2 (2nd dose)	2	2.0 ± 2.8 (0–4)	2 (100%)	17.4 ± 4.9	1 (50.0%)	50.7 ± 30.5	0	7.0 ± 5.9
Incomplete vaccination (N = 85)								
mRNA-1273 (1st dose)	62	9.9 ± 4.0 (4–18)	35 (56.5%)	54.5 ± 316.8	36 (58.1%)	1925.6 ± 8661.8	33 (53.2%)	35.4 ± 39.2
ChAdOx1 (1st dose)	22	66.2 ± 31.3 (7–117)	22 (100%)	90.1 ± 212.6	20 (91.0%)	409.7 ± 365.2	15 (68.2%)	39.7 ± 24.0
BNT162b2 (1st dose)	1	8	0	0.4	0	9.6	0	11.7

* Mean ± SD.

**Table 3 vaccines-10-00193-t003:** Comparison of antibody quantities between the adverse event group and no discomfort group ≥15 days after the second vaccination of BNT162b2.

Variable	Mean ± SD		*p* Value
	Adverse Event Group	No Discomfort Group	
ChAdOx1 (*n* = 277)			
Number	174	103	
Age	41.9 ± 10.9	47.2 ± 10.9	<0.01
Days after the last vaccination	30.7 ± 8.0	29.1 ± 7.7	NS
Elecsys Anti-SARS-CoV-2 (U/mL)	1037.1 ± 708.8	1051.6 ± 733.8	NS
Alinity SARS-CoV-2 IgG (AU/mL)	1817.8 ± 2175.4	1718.7 ± 2365.0	NS
R-FIND SARS-CoV-2 Neutralizing Antibody (%)	82.3 ± 19.6	83.0 ± 15.8	NS
BNT162b2 (*n* = 83)			
Number	46	37	
Age	35.4 ± 11.5	41.0 ± 13.7	NS
Days after the last vaccination	94.5 ± 1.9	94.9 ± 2.5	NS
Elecsys Anti-SARS-CoV-2 (U/mL)	1065.8 ± 611.6	871.2 ± 392.7	NS
Alinity SARS-CoV-2 IgG (AU/mL)	4118.0 ± 2736.8	3111.8 ± 1538.3	<0.05
R-FIND SARS-CoV-2 Neutralizing Antibody (%)	92.2 ± 5.6	90.4 ± 7.2	NS

## Data Availability

All the data available is provided in this paper.

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
