# Peer review of "Comparing SARS-CoV-2 Antibody Responses after Various COVID-19 Vaccinations in Healthcare Workers"

_vaccines, 2022, doi:10.3390/vaccines10020193_

Round 1
Reviewer 1 Report
In the current manuscript, the authors analyzed SARS-COV-2 antibody using various types of assays. The experiments are mostly well performed, and the manuscript is almost written well. Here are my comments that would improve the manuscript.
- Detailed method about the period from blood collection and the assays is missing. If the authors used the already stored samples in the current study, did the author check the effect of cryopreservation?
- It is unclear about “No response” in Table 1 though I assume that it shows “not available”.
- In Table 3, adjustments of the age at the vaccination, gender and duration from the vaccination are needed to compare the titers.
- The rationale about the incidence of adverse effects is unclear in the current manuscript. It may be meaning to compare the relationships between incidence and the results of assays conducted in the current study.
Author Response
In the current manuscript, the authors analyzed SARS-COV-2 antibody using various types of assays. The experiments are mostly well performed, and the manuscript is almost written well. Here are my comments that would improve the manuscript.
1. Detailed method about the period from blood collection and the assays is missing. If the authors used the already stored samples in the current study, did the author check the effect of cryopreservation?
- After blood collection, the serum was kept refrigerated, and antibody tests were performed within 3 days. (page 2, line 51-52)
2. It is unclear about “No response” in Table 1 though I assume that it shows “not available”.
- We edited follow your comment.
3. In Table 3, adjustments of the age at the vaccination, gender and duration from the vaccination are needed to compare the titers.
- We performed statistics again to adjust the age, gender, and duration after vaccination. And we concluded that the differences in duration between the two groups were too prominent to assess differences between vaccine types. Therefore, we deleted the comparison between the two vaccine groups, including Table 3 in this paper.
4. The rationale about the incidence of adverse effects is unclear in the current manuscript. It may be meaning to compare the relationships between incidence and the results of assays conducted in the current study.
- The content could not be included because there were no significant results from the analysis of side effects and antibody values. We agree with your opinion, and the content has been deleted from the abstract.
Thank you for reviewing and commenting on our manuscript.
Reviewer 2 Report
Dear Editors and Authors,
I have carefully read your manuscript entitled "Comparing SARS-CoV-2 antibody responses after various COVID-19 vaccinations in healthcare workers". In my opinion the paper is valuable and interesting. In the pandemic time it is very important to publish every statistics and comparisons about virus and vaccinations. Only a multidisciplinary and broad perspective will allow us to control the situation. I suggest minor corrections.
My comments:
- precise the title - add "(South Korea)" to the and of the sentence;
- consider keywords different than words in the title;
- extend the Introduction section with the latest literature information on the research and statistics that you have found so far; introduce the reader to the topic;
- Figure 1 - "Days atter" or maybe "days after"?
- Figure 2 - Do these charts show the percentage? Maybe it would be worth adding specific values on individual parts of the chart?
Best regards,
Reviewer
Author Response
- precise the title - add "(South Korea)" to the and of the sentence;
- We edited follow your comment.
- consider keywords different than words in the title;
- We edited follow your comment.
- extend the Introduction section with the latest literature information on the research and statistics that you have found so far; introduce the reader to the topic;
- Based on your comments, we have supplemented the text and added references.
- Figure 1 - "Days atter" or maybe "days after"?
- We edited follow your comment.
- Figure 2 - Do these charts show the percentage? Maybe it would be worth adding specific values on individual parts of the chart?
- We edited follow your comment.
Thank you for reviewing and commenting on our manuscript.
Round 2
Reviewer 1 Report
The authors responded the reviewer's comments and modified the manuscript. The changes have made the manuscript clearer. From the response to #3, I understand that the authors analyzed the relationship between the titers and incidence of adverse effects. Regardless of the significance, probably from just statistical point, this point will provide novel features about the vaccination. Thus, the authors need to show the important results in the manuscript.
Author Response
The authors responded the reviewer's comments and modified the manuscript. The changes have made the manuscript clearer. From the response to #3, I understand that the authors analyzed the relationship between the titers and incidence of adverse effects. Regardless of the significance, probably from just statistical point, this point will provide novel features about the vaccination. Thus, the authors need to show the important results in the manuscript.
- We performed statistics again on the relationship between adverse events and antibody quantities. Thanks to your advice, we obtained significant results and added the contents to the manuscript (blue-colored sentences: line 173-187, table 3, line214-224, and others).
Thank you for reviewing and commenting on our manuscript.